# Olefin-Metathesis-Derived Norbornene–Ethylene–Vinyl Acetate/Vinyl Alcohol Multiblock Copolymers: Impact of the Copolymer Structure on the Gas Permeation Properties

**DOI:** 10.3390/polym14030444

**Published:** 2022-01-22

**Authors:** Alexey V. Roenko, Roman Y. Nikiforov, Maria L. Gringolts, Nikolay A. Belov, Yulia I. Denisova, Georgiy A. Shandryuk, Galina N. Bondarenko, Yaroslav V. Kudryavtsev, Eugene S. Finkelshtein

**Affiliations:** A.V. Topchiev Institute of Petrochemical Synthesis, Russian Academy of Sciences, Leninskii pr. 29, 119991 Moscow, Russia; royenko@ips.ac.ru (A.V.R.); nru@ips.ac.ru (R.Y.N.); belov@ips.ac.ru (N.A.B.); denisova@ips.ac.ru (Y.I.D.); gosha@ips.ac.ru (G.A.S.); bond@ips.ac.ru (G.N.B.); fin314@gmail.com (E.S.F.)

**Keywords:** metathesis polynorbornene, ethylene–vinyl acetate/vinyl alcohol copolymers, multiblock copolymers, post-modification, hydrogenation, epoxidation, gas barrier properties, permeability, diffusivity

## Abstract

Commercial metathesis polynorbornene is used for the fabrication of high-damping coatings and bulk materials that dissipate vibration and impact energies. Functionalization of this non-polar polymer can improve its adhesive, gas barrier, and other properties, thereby potentially expanding its application area. With this aim, the post-modification of polynorbornene was carried out by inserting ethylene–vinyl acetate–vinyl alcohol blocks into its backbone via the cross-metathesis of polynorbornene with poly(5-acetoxy-1-octenylene) and subsequent deacetylation and hydrogenation of the obtained multiblock copolymers. For the first time, epoxy groups were introduced into the main chains of these copolymers, followed by the oxirane ring opening reaction. The influence of post-modification on the thermal, gas separation, and mechanical properties of the new copolymers was studied. It was shown that the gas permeability of the copolymer significantly depends on its composition, as well as on the amounts of hydroxyl and epoxy groups. The developed methods efficiently improve the barrier properties, reducing the oxygen permeability by 15–33 times in comparison with polynorbornene. The obtained results are promising for various applications and can be extended to a broader family of polydienes and other polymers containing backbone double bonds.

## 1. Introduction

The development of new polymer materials heavily relies on the possibility to impart new properties to known polymers. Polynorbornenes are recognized as convenient objects for exploratory research in membrane materials science [1,2]. Their use provides numerous opportunities for the macromolecular design of membrane materials and the establishment of structure–property relationships. The main approaches in the chemistry of polynorbornenes include: (i) tailoring the repeating unit at the stage of monomer synthesis by introducing the desired substituents (most often Si- or F-containing bulky groups) or choosing the type of norbornene-containing fragment (norbornene, norbornadiene, tricyclo[4.2.1.0^2,5^]non-7-ene, tricyclo[4.2.1.0^2,5^]non-3,7-diene, etc.); (ii) the design of the monomer unit sequence via polymerization according to the metathesis (ROMP), addition (AP), or catalytic arene–norbornene annulation (CANAL) schemes. Addition and CANAL polynorbornenes lead to rigid polymer chains, which can demonstrate high gas separation parameters, in some cases beyond the Robeson upper bound [2]. Metathesis polynorbornenes, as a rule, are less permeable and less stable chemically and thermally due to the presence of double bonds in their backbone. Recently, the post-modification of metathesis polynorbornenes was suggested to adjust their gas separation properties and improve their thermochemical stability [2,3,4,5,6,7,8,9,10]. The proposed modification routes included hydrogenation [4,11], epoxidation [3], *gem*-difluorocyclopropanation [5], and cross-linking [6,7,8,9,10] of the double bonds. It was shown that the epoxidation led to a significant decrease in gas permeability of metathesis polynorbornenes due decreases in diffusion and solubility coefficients for almost all gases [3]. The only exception was the CO_2_ solubility coefficient, which increased upon introducing oxirane rings into the polynorbornene backbone. *Gem*-difluorocyclopropanation led to higher gas permeability of glassy polynorbornenes, whereas the ideal selectivity remained practically unchanged and even increased for gas pairs that included methane [5]. An efficient approach was used to obtain copolymers containing the fragments of polysiloxane and substituted polynorbornenes, cross-linked in situ through norbornene units [6,7,8,9,10]. Some of these copolymers demonstrated the gas permeability values above the Robeson upper bound [7,9,10]. Recently, we developed a method for the post-modification of metathesis polynorbornenes by involving them in a macromolecular cross-metathesis reaction with linear polyenes [12,13]. The process is an interchain exchange reaction, during the course of which different macromolecules reshuffle their segments via the rupture and creation of double bonds in accordance with the olefin metathesis mechanism. The initial exchange of segments between the parent homopolymers yields a diblock copolymer, and later a statistical multiblock copolymer is formed. The average block lengths gradually decrease with time and finally reach the values typical of a copolymer with a completely random sequence of monomer units [12,13,14].

The interchain reaction of macromolecular cross-metathesis allows the insertion of chemically different polyene fragments into the polynorbornene backbone. In this way the copolymers of multiblock structure were obtained from polynorbornene and polyoctenamer or its functional derivatives [14,15,16,17]. It is worth noting that block copolymers containing norbornene segments can exhibit interesting gas separation properties. Mainly, these are di-, tri-, and tetrablock copolymers of the vinyl-addition or metathesis type, with various substituents and affinities for certain gases, or with different ionic conductivity levels [18,19]. At the same time, the gas separation properties of statistical multiblock copolymers have hardly been studied. We have recently shown that new multiblock norbornene–ethylene–vinyl acetate/vinyl alcohol copolymers can be synthesized via the macromolecular cross-metathesis of polynorbornene with poly(5-acetoxy-1-octenylene) followed by hydrogenation and deacetylation [20]. Interest in such copolymers is associated with the possibility of obtaining materials that combine the properties of original homopolymers. The well-known ethylene–vinyl acetate copolymers (Et–VOAc) and ethylene–vinyl alcohol copolymers (Et–VOH) are widely applied as adhesives, coatings, foams, water bottles, and gas barrier materials, as well as in plumbing [21]. Metathesis polynorbornene, produced under the Norsorex^®^ trademark, is used for the manufacture of coatings and materials with high grip performance and outstanding vibration and impact damping capability [22,23].

This study is based on the idea that functionalization of nonpolar polynorbornene with blocks of Et–VOAc and Et–VOH copolymers can expand the area of its possible application, and in particular can improve the properties of polynorbornene coatings by increasing their gas barrier characteristics. It is worth noting that polynorbornenes were used as matrices for oxygen barrier materials [24,25]. In this paper, we begin with the synthesis of multiblock copolymers from polynorbornene and poly(5-acetoxy-1-octenylene) via the cross-metathesis reaction. Then, we modify them by hydrogenation, epoxidation, and deacetylation to obtain a series of norbornene–ethylene–vinyl acetate/vinyl alcohol multiblock copolymers with different chain structures and substituents. The copolymer characterization is focused on measuring the gas permeability parameters and discussing their relations with the copolymer structure and method of modification.

## 2. Materials and Methods

### 2.1. Materials

All manipulations involving air- and moisture-sensitive compounds were carried out in oven-dried glassware using dry solvents and standard Schlenk and vacuum-line techniques under an argon atmosphere. The 5-acetoxy-1-cyclooctene (COAc) monomer was synthesized according to the procedure described in [26]. Norbornene and cis, cis-1,5-cyclooctadiene (Sigma-Aldrich, St. Louis, MO, USA) were dried over sodium and stored under argon. Norbornene was used as a 4.6 M solution in dry chloroform. Grubbs’ catalysts of the 1st (Cl_2_(PCy_3_)_2_Ru=CHPh, G1) and 2nd (Cl_2_(PCy_3_)ImesRu=CHPh, G2) generations (Sigma-Aldrich, St. Louis, MO, USA) were used without purification in the form of 0.002–0.022 M solutions in dry chloroform. m-Chloroperbenzoic acid, p-toluenesulfonyl hydrazide, and inhibitor of oxidation 2,2′-methylene-bis (6-tert-butyl-4-methylphenol) (Sigma-Aldrich, St. Louis, MO, USA) were used as supplied. Other reagents and solvents were purified using common procedures.

### 2.2. Characterization

Nuclear magnetic resonance (NMR) measurements were carried out at room temperature using a Bruker (Billerica, MA, USA) Avance III HD (400 MHz) spectrometer at 400.1 MHz (^1^H NMR) and 100.6 MHz (^13^C NMR) in CDCl_3_ or DMSO-d6 solutions. ^1^H and ^13^C chemical shifts in ppm relative to TMS were determined using the signals of residual CHCl_3_ (7.28 ppm) and CDCl_3_ (77.23 ppm), respectively.

The molar mass distribution of the polymers was determined by gel permeation chromatography (GPC) on a modular high-pressure chromatograph equipped with a LabAlliance Series 1500 constant flow pump (Scientific Systems, Woburn, MA, USA), refractometric LKB Bromma RI 2142 detector, and serially connected Waters WAT054460 and Tosoh Biosep G3000HHR columns, with THF as the solvent, a flow rate of 1 mL min^−1^, sample volume of 100 μL, and sample concentration of 1 mg L^−1^. The weight-average molar mass *M*_w_ and dispersity *Đ* were calculated using a standard procedure relative to polystyrene standards (Agilent, Santa Clara, CA, USA).

Differential scanning calorimetry (DSC) thermograms were recorded on a Mettler TA 4000 (Greifensee, Switzerland) system at a rate of 10 °C min^−1^ under an argon flow of 70 mL min^−1^ in the range of −100 °C to 100 °C. The data were processed using the STARe (Mettler Toledo, Columbus, OH, USA) service program. The accuracy levels of measurements were 0.3 °C for temperature and 1 J g^−1^ for enthalpy.

FTIR spectra were recorded on a Bruker (Billerica, MA, USA) IFS 66v/S FTIR spectrometer in transmission mode in the range of 400–4000 cm^−1^ (50 scans, resolution of 2 cm^−1^) at room temperature. To treat the experimental data, the OPUS 7 software package (Bruker) was used.

An I1140M-5-01-1 universal tensile testing machine (Tochpribor-KB, Ivanovo, Russia) and the ASTM D638 method were used for mechanical testing of the films. Dog bone tensile specimens were prepared by punching the cast films using a stainless steel die (ASTM D1708-96, 22 × 5 mm^2^, a thickness of 20–40 µm) at room temperature.

Permeability (*P*) and diffusion (*D*) coefficients were measured using a MKS Baratron (Andover, MA, USA) capacitance manometer-based setup [3]. The experiments were performed at 22 ± 2 °C, with an upstream pressure of 760 ± 10 Torr and downstream pressure of 0–5 Torr. The Daynes–Barrer (time-lag) method was applied for the estimation of *D* [27]: *D* = *l*^2^/6*θ*, where *l* is the thickness of the tested film and *θ* is the so-called time lag. The solubility coefficients of gases were estimated as a ratio of *P/D*. The measurements of the steady-state gas flux across the film were performed at times *t* > (4–6) *θ*. For each copolymer, at least two samples were tested, and typical errors of *P* and *D* measurements did not exceed 10 and 20%, respectively.

### 2.3. Polymer Synthesis and Modification

Polynorbornene (PN) was synthesized by ROMP according to the procedure described in [28].

Poly(5-acetoxy-1-octenylene) (PCOAc) (Appendix A) was synthesized by ROMP according to the procedure described in [20,29].

Cross-metathesis between PN and PCOAc was accomplished as described in [20]. Immediately before the cross-metathesis, the initial homopolymers were purified from catalyst and inhibitor residues by passing their solutions in chloroform (12% wt for PCOAc and 2% wt for PN) through a column of silica gel with chloroform as the eluent and by precipitating the purified polymer into ethanol. The polymers were thoroughly dried under reduced pressure to a constant mass for at least 2 days. The following procedure describes the synthesis of the (N–COAc)C2 copolymer and corresponds to the molar ratio [PN]:[PCOAc]:[G2] = 2400:2650:1 and the mass ratio [PN]:[PCOAc] = 1:2. The mixture of PN (1 g, 10.8 mmol) and PCOAc (2 g, 12 mmol) was placed in a round-bottom two-neck flask, then dry chloroform (19 mL) was added under argon and the system was left overnight. Then, it was degassed three times using the freeze–pump–thaw cycle technique before 1 mL (0.0044 mmol) of G2 solution in chloroform was added. The cross-metathesis was carried out under argon and stopped after a required time by adding ethyl vinyl ether at a molar-to-catalyst ratio of 500:1 and stirring the reaction mixture for 30 min. Then, the oxidation inhibitor was added, and after stirring for 0.5–1 h, the copolymer was precipitated into ethanol and dried under reduced pressure to a constant mass. The (N–COAc)C2 yield was 2.4 g (80%). ^1^H and ^13^C spectra of (N–COAc) copolymers are shown in Appendix A.

Hydrogenation of (N–COAc)C2 was carried out as described in [20]. The copolymer (0.67 g) was dissolved in dry o-xylene (60 mL) under an argon atmosphere. Then, p-toluenesulfonyl hydrazide (TsH) (4.42 g, 0.0238 mol) and the oxidation inhibitor (30 mg, 10% of the polymer mass) were added under vigorous stirring. The mixture was degassed using three freeze–pump–thaw cycles, stirred for 5.5 h at reflux, and slowly poured into an excess of methanol. The solid polymer was collected by filtration, washed twice with pure methanol, and dried under reduced pressure to a constant mass. The yield of the saturated H(N–COAc)C2 copolymer was 0.46 g (67%). Its ^1^H and ^13^C spectra are shown in Appendix A.

Epoxidation of (N–COAc)C copolymers was carried out according to the previously developed technique [30,31] to minimize the loss in molar mass. The method is given for the epoxidation of C2. A solution of (N–COAc)C2 (1.1 g, 0.0084 mol C=C bonds) in 63 mL of absolute toluene containing the oxidation inhibitor (5% wt) was prepared for 1.5 h under argon in a two-necked reactor (*V* = 25 mL) equipped with a magnetic stirrer. Then, 77% m-chloroperbenzoic acid (3.77 g, 0.0169 mol of 100% peracid) was added to the reaction mixture (yellowish suspension) at 15 °C and stirred for 1.5 h. Then, the polymer was precipitated into methanol and dried in vacuum until constant mass to isolate 0.97 g (80%) of epoxidized E(N–COAc)C2 copolymer. ^1^H and ^13^C NMR spectra of E(N–COAc)C are shown in Appendix A.

Deprotection of Ac groups was carried out as described in [20]. A E(N–COAc)C1 copolymer (0.494 g, 0.000903 mol of AcO-groups) solution in dry THF (35 mL) was prepared under an argon atmosphere and degassed three times. Separately, a 0.9–1.0 M solution of sodium methoxide in methanol (usually 0.093 g Na was added to 5 mL MeOH) was prepared under argon. A 3.9 mL 1 M sodium methoxide solution (0.00398 mol NaOMe) at 0 °C was added dropwise to the copolymer solution, which became cloudy and then transparent. The mixture was stirred at room temperature for 5 h, then concentrated in vacuum and poured into slightly acidic methanol (0.75 mL 1 M HCl, 25 mL MeOH). The isolated polymer was washed three times with pure methanol and thoroughly dried under reduced pressure to a constant mass for at least two days. The yield of the E(N–COH)C1 copolymer was 0.385 g (85% mol based on OH-groups). It was soluble in CHCl_3_/MeOH (5:1 vol) mixture at room temperature.

The same procedure was used for the deprotection of Ac-groups in H(N–COAc)C2. The saturated H(N–COH)C2 copolymer was precipitated in the reaction mixture during deacetylation. To isolate the product, it was washed with acidified ethanol, then with pure ethanol and hexane. H(N–COH)C2 was soluble in THF or chloroform–methanol mixture only when heated above 55 °C.

A copolymer E(N–COAc)C2 (0.8121 g, 0.0027 mol of AcO-groups) solution in dry THF (60 mL) was prepared under an argon atmosphere and degassed three times. Separately, a 0.89 M solution of sodium methoxide in methanol (usually 0.21 g Na was added to 10.3 mL MeOH) was prepared under argon. A 7.3 mL 0.89 M sodium methoxide solution (0.0065 mol NaOMe) was added dropwise at 0 °C to the copolymer solution, which became cloudy and then transparent. The mixture was stirred at room temperature for 5 h and then concentrated in a vacuum to 1/3 of the reaction mixture volume. A 2M HCl solution was added dropwise to the concentrate until pH = 7. The neutralized mixture was poured in hexane. The isolated polymer was washed three times with pure hexane and thoroughly dried under reduced pressure to a constant mass for at least two days. The yield of the E(N–COH)C2 copolymer was 0.4913 g (70% mol based on HO-groups). The copolymer was insoluble in CHCl_3_ even after heating, was poorly soluble in CHCl_3_/MeOH (5:1 vol) mixture at 55 °C, and was soluble in THF at 55 °C, as well as in DMSO at room temperature. ^1^H NMR spectra of copolymers E(N–COH)C are shown in Appendix A.

Films with a thickness range of 20–30 μm were cast from a filtered 3% copolymer solution onto cellophane. After solvent evaporation, the films were dried under vacuum at room temperature to a constant mass. Solvents (chloroform, its mixture with methanol, and THF) and the casting temperature (room temperature or 55–60 °C) were chosen depending on the solubility of copolymers: CHCl_3_ for (N–COAc)C1 and E(N–COAc)C1; CHCl_3_/MeOH (5:1 vol) for E(N–COH)C1, THF at 55 °C for E(N–COH)C2. A film of the H(N–COH)C2 copolymer was cast from 3% wt solution in a mixture of toluene with isopropanol (5:1 vol) onto a silanized glass surface heated to 55 °C. The film was dried at 60 °C for 6 h, then in vacuum at room temperature to a constant mass.

## 3. Results and Discussion

### 3.1. Polymer Synthesis

The initial homopolymers, namely polynorbornene (PN) and poly(5-acetoxy-1-octenylene) (PCOAc), were synthesized by ring-opening metathesis polymerization (ROMP) of norbornene and 5-acetoxy-1-cyclooctene (COAc) in the presence of 1st (G1)- and 2nd (G2)-generation Grubbs’ catalysts (Figure 1). Highly strained norbornene was readily polymerized using the G1 catalyst as described in [28]. A more active G2 catalyst was needed to obtain a high molar mass PCOAc by ROMP of low-strained COAc [20]. The PN was a glassy amorphous polymer with predominantly trans-C=C double bonds (80%), whereas PCOAc (trans-C=C 84%) was a rubbery amorphous polymer.

The cross-metathesis reaction between homopolymers, namely macromolecular cross-metathesis (MCM), was accomplished using various weight ratios of homopolymers to obtain (N–COAc)C statistical multiblock copolymers with different backbone structures (Figure 1 and Table 1). The reaction was carried out in a chloroform solution with sufficiently high polymer concentration, which facilitated MCM [13]. The catalyst solution was added to the polymer mixture. After a required time period, the reaction was stopped by adding ethyl vinyl ether and the copolymers were precipitated into alcohol and dried in a vacuum. An inhibitor was added to the polymer solutions before polymer precipitation. During MCM, the catalyst cleaved homopolymer chains according to the olefin metathesis mechanism and formed Ru–carbene macromolecular complexes. The reshuffling of PN and PCOAc segments resulted in the formation of (N–COAc)C multiblock copolymers. Their composition and average block lengths were determined by ^13^C NMR spectroscopy according to the previously developed technique (see SM) [20]. The appearance of signals from carbon atoms of heterodyads (c and d on Figure 1) in the ^13^C NMR spectrum indicates the formation of (N–COAc)C (C_c,_ C_d_ in Figure 1).

We obtained two samples of rubbery (N–COAc)C: C1 (*T*_g_ = −1 °C) with a three-fold excess of norbornene units and C2 (*T*_g_ = −29 °C) with an equimolar ratio of norbornene to COAc units and rather long block lengths (Table 1 and Figure 1).

### 3.2. Copolymer Post-Modification

To increase the thermochemical stability and enhance the barrier properties of (N–COAc)C copolymers, they were modified by hydrogenation, epoxidation, and deacetylation. Hydrogenation and epoxidation of double bonds in polynorbornenes improve their stability during storage [30,32,33,34]. Additionally, epoxidation improves the barrier properties of PN by significantly reducing its gas permeability [3]. It is also known that replacing the acetoxy substituent with the HO-group in ethylene–vinyl acetate copolymers improves their barrier properties [21,35]. In this regard, we chose two methods of copolymer modification (Figure 2). The first one included epoxidation of the both (N–COAc)C copolymers to obtain E(N–COAc)C and subsequent deacetylation to prepare hydroxyl-containing E(N–COH)C multiblock copolymers. For the second method, the (N–COAc)C2 copolymer, which contained more COAc units than the (N–COAc)C1 copolymer, was hydrogenated to H(N–COAc)C2 and then deacetylated to H(N–COH)C2, thereby introducing ethylene–vinyl alcohol blocks into PN.

Epoxidation of (N–COAc)C was carried out with m-chloroperbenzoic acid (mCPBA) (Figure 2) in a toluene solution according to the published procedure [4,5]. Upon epoxidation, the ^1^H NMR spectrum contained the signals of epoxy ring protons at 2.76 and 2.63 ppm (Figure 2B), whereas the C=C double bond proton signals at 5.34 and 5.20 ppm (Figure 2A) disappeared. This means that the oxirane rings completely replaced double bonds under the chosen conditions. Despite the oxidation inhibitor presence (5% wt) in the reaction mixture and the minimum reaction time (1.5 h) required for almost complete conversion of C=C bonds, the molar mass of the product decreased by 1.7–2.5 times (Table 2, rows 1 and 2; Appendix A). The copolymer (N–COAc)C2, which contained more COAc blocks, degraded less than the copolymer C1 with predominantly norbornene blocks. Epoxidation increased the glass transition temperature of the copolymers, C1 turned into a glassy copolymer, while C2 remained an elastomer. The both epoxidized copolymers were amorphous and easily soluble in chloroform.

Hydrogenation of (N–COAc)C2 led to the H(N–COAc)C2 copolymer, consisting of polynorbornene (HPN) and poly(5-acetoxy-1-cyclooctenylene) (HPCOAc) blocks without double bonds (Figure 2). Note that the hydrogenated HPCOAc homopolymer is structurally similar to the commercial ethylene–vinyl acetate copolymer (Et–VOAc). The peculiarity of HPCOAc lies in the regularly located AcO-groups strictly at every eighth carbon atom. In the present study, the copolymer (N–COAc)C2 with higher content of COAc units was hydrogenated because one could expect a more pronounced effect of introducing Et–VOAc blocks on the gas permeability. The reaction was carried out in o-xylene in the presence of p-toluenesulfonylhydrazide (TsH) (Figure 2) under the conditions specified in [20]. According to ^1^H NMR spectrum, the resulting copolymer, H(N–COAc)C2, did not contain double bonds (Appendix A). It is known that after the reduction of double bonds, polynorbornene (HPN) acquires crystallinity (*T*_m_ = 144 °C), loses its solubility in chloroform and toluene, and its *T*_g_ decreases from 39 to −13 °C [36]. However, the saturated copolymer H(N–COAc)C2 did not lose its solubility in chloroform and remained amorphous due to the presence of HPCOAc blocks (Table 2, row 3). Additionally, the decrease in *T*_g_ of C2 (from −29 to −36 °C) was less pronounced than for HPN.

The deprotection of acetyl groups in copolymers was carried out in the presence of sodium methoxide in a THF solution (Figure 2) [20,37,38]. Upon deacetylation of the epoxidized norbornene-enriched copolymer E(N–COAc)C1, the ^1^H NMR spectrum no longer contained the signals of protons in the acetyl group (2 ppm) or in the nearest –CH-group of the backbone (4.8 ppm) (Figure 2C), while the ^13^C NMR spectrum did not reveal the carbonyl signal (170.9 ppm, Appendix A). At the same time, the signals of the epoxy group protons (2.61–2.87 ppm, Appendix A) were retained. These changes were confirmed by FTIR spectroscopy: a (C=O) band at 1737 cm^−1^ disappeared; an (O–H) band at 3476 cm^−1^ appeared; and (epoxy) bands at 1240 cm^−1^ (coinciding with the acetyl group band), 888 cm^−1^, and 804 cm^−1^ were kept intact (Appendix A). A broad (O–H) band at 3476 cm^−1^ indicated association [39]. A new 1045 cm^−1^ band, which appeared on the shoulder of the 1067 cm^−1^ band, is usually related to the presence of ether O–C–O bonds. This could be explained by the formation of interchain cross-links through the oxygen atoms upon epoxide ring opening. As a result of deacetylation, the molar mass of the copolymer decreased slightly (Table 2, rows 1,4), while the polymer became poorly soluble in chloroform and dissolved in a chloroform–methanol mixture at room temperature (Table 2, row 4).

In contrast to C1, the deacetylated C2 copolymer did not precipitate into ethanol from the reaction mixture. In order to isolate it, the reaction mixture neutralized by HCl was precipitated into hexane. The ^13^C NMR spectrum of the E(N–COH)C2 copolymer in the CDCl_3_–DMSO-d6 (1:1 vol) mixture contained signals of the carbon atom linked to the HO-group (66–74 ppm, Appendix A, similarly to [35]), whereas practically no signals of the carbon atoms in the epoxy group (56–62 ppm, Appendix A) were detected. New signals at 81.6 and 82.0 ppm (Appendix A) were attributed to the carbon atoms in C–O–C, including C–O–CH_3_ [40], formed as a result of the epoxy ring opening by sodium methylate in the presence of HCl. The same weak signals were observed in the ^13^C NMR spectrum of E(N–COH)C1 (Appendix A). Deacetylation of E(N–COAc)C2 was confirmed by the absence of the Ac-group bands in the FTIR spectrum (Appendix A). In that case, more hydroxyl groups were formed compared with the hydrogenated copolymer H(N–COH)C2 with the same initial content of Ac-groups (according to the relative intensities of 3391 cm^−^^1^ and 2940 cm^−1^ bands from CH_2_). This indicates the formation of hydroxyl groups due to the epoxide groups opening (Appendix A). In addition, the IR spectrum of E(N–COH)C2 (Appendix A) reveals an upshift in the bands related to the HO-groups (3476 to 3391 cm^−^^1^ and 1067 to 1053 cm^−^^1^) and O–C–O bonds (1045 to 1032 cm^−^^1^), which is a direct indication of hydrogen bonding between hydroxyls and ether oxygen atoms [35,39]. The formation of C–O–C bonds is indicated by a significant increase in the intensity of the bands in the 1100–1000 cm^−^^1^ region. The relative content of epoxy groups in E(N–COAc)C1, E(N–COH)C1, and E(N–COH)C2 copolymers can be estimated from the intensity of the epoxy band (888 cm^−^^1^) normalized to the CH_2_ band intensities (2940 and 1461 cm^−^^1^). Taking into account the copolymer composition, both C1 copolymers contain approximately the same amount of epoxy groups, and C2 contains 6 times less of these groups (Appendix A). The molar mass of C2 after deacetylation almost doubled (Table 2, rows 2,5), which may indicate the presence of cross-links in the copolymer. According to the DSC curves, the epoxidized C1 and C2 copolymers remained amorphous, while *T*_g_ increased by 14 degrees for C1 (from 31 to 45 °C, Table 2, rows 1,4) and by 84 degrees for C2 (from 6 to 90 °C, Table 2, rows 2,5). This can be explained by the higher content of hydroxyls in C2, including those formed due to the decomposition of oxirane rings, which promotes physical cross-linking of the copolymer with hydrogen bonds (Figure 2). This agrees with the lower solubility of deacetylated C2 relative to C1, namely E(N–COH)C2 is soluble in a chloroform–methanol mixture only upon heating (Table 2, row 5). At the same time, the expected C–O–C cross-links in the deacetylated E(N–COH)C2 copolymer did not prevent its solubility in DMSO.

Saturated H(N–COH)C2 copolymer was precipitated in the reaction mixture during deacetylation. To isolate the product, it was washed with acidified ethanol, then with pure ethanol and hexane. The copolymer obtained was soluble in THF or chloroform–methanol mixture only when heated above 55 °C. Its FTIR spectrum (Appendix A) did not contain any spectral signs of the acetyl group or unsaturation. The bands of the alcohol group were strongly upshifted toward 3317cm^−^^1^. New bands appeared in the region of 1130–1140 cm^−^^1^, where rather intense crystallinity bands of PVOH usually lie [41]. According to DSC (Table 2, row 6), the removal of acetyl groups in the hydrogenated C2 led to an increase in *T*_g_ and to the appearance of crystallinity.

All synthesized copolymers were amorphous, with the exception of H(N–COH)C2 (Table 2, row 6; Appendix A), which was characterized by *T*_g_ = 29 °C, *T*_m_ = 120 °C and Δ*H* = 30 J·g^−^^1^. Taking into account the copolymer composition ([HN]:[HCOH] = 2:3 by mass), the enthalpy of fusion is equivalent to 50 J per gram of HCOH blocks or 75 J per gram of HN blocks. The melting peak on the thermogram (Appendix A, curve 2) is unimodal and not very wide, meaning it is reasonable to assume that crystals of a single type are formed. Since H(N–COAc)C2 is amorphous, we can suppose that they are built of vinyl alcohol and ethylene units. Indeed, for a Et–VOH copolymer with a VOH content of 25 mol%, [20] reports *T*_g_ = 35 °C, *T*_m_ = 126 °C, and Δ*H* = 50 J·g^−^^1^, which are rather close to the above values for H(N–COH)C2. For hydrogenated PN, *T*_m_ is higher, falling within the range of 136–160 °C, which depends on the hydrogenation method and type of catalyst [36]. In this study, we used the stoichiometric diimide hydrogenation to obtain HPN with *T*_g_ = −11 °C, *T*_m_ = 143 °C, and Δ*H* = 55 J⋅g^−^^1^, which are close to the data in [42] of *T*_m_ = 140.8 °C and Δ*H* = 58.7 J⋅g^−^^1^ but rather far from the values for H(N–COH)C2.

We know from the literature [43] that Et–VOH copolymers with 25% mol of vinyl alcohol units form crystals that are transitional from orthorhombic PE to monoclinic PVOH; therefore, it not easy to estimate the degree of crystallinity from the DSC data. Taking into account that the specific enthalpy of fusion is equal to 293 J⋅g^−1^ and 161 J⋅g^−1^ for 100% crystalline PE and PVOH, respectively, we can only conclude that the degree of crystallinity of H(N–COH)C2 is in the range of 50/293 ≈ 0.17 to 50/161 ≈ 0.31.

More information on the crystallinity could be obtained with X-ray scattering, but this requires an extensive synthetic study to obtain the H(N–COH)C2 copolymers of various compositions and degrees of blockiness. Here, we simply refer to the standard XRD patterns of H(N–COAc)C2, E(N–COH)C2, and E(N–COH)C1 copolymers shown in Appendix A. Whereas both epoxy-containing copolymers are amorphous, the hydrogenated sample reveals crystalline peaks. The positions of the two main peaks located closely around 2θ = 20° give some evidence that we are dealing with the crystallinity of PVOH rather than PE, where the main peak is expected at 2θ > 21° [44].

### 3.3. Gas Permeation and Other Properties

This study implements several methods of post-modification intended to improve the gas barrier properties of metathesis polynorbornene: the introduction of COAc fragments, the epoxidation of double bonds in the copolymers of norbornene with COAc, the hydrogenation of double bonds, and the deacetylation of AcO-groups. The molar ratio of norbornene to substituted COAc units in the (N–COAc)C1 copolymer corresponds to 3.3:1 and the average block lengths are equal to 10 and 3, respectively. The other physicochemical parameters are presented in Table 3. The block length ratio was chosen to adjust the copolymer *T*_g_ so that the film preparation and processing became more convenient. For (N–COAc)C2, the equimolar 1:1 ratio of norbornene to COAc units was used.

The monomer unit of a metathesis PCOAc homopolymer based on COAc approximately contains one vinyl acetate (VOAc) per three ethylene (Et) units and corresponds to a 75/25 (mol%) Et–VOAc copolymer. The pure poly(vinyl acetate) (PVOAc) is distinguished by a low level of gas permeability (*P*(O_2_) = 0.3–0.5 Barrer, Table 4) and a *T*_g_ close to 35 °C. The introduction of ethylene units to its structure gradually lowers the *T_g_* of copolymers through −5 (12 mol% of Et) [45] and −26 (60 mol% of Et) to −125 °C (pure polyethylene) [46]. One can assume that the backbone flexibility in Et–VOAc copolymers grows with an increase in the number of ethylene fragments. As seen from Table 4, which presents the literature data, this results in a dramatic growth of the diffusivity and permeability of gases; an 8–9-fold increase in permeability can be observed for the copolymer with 12 mol% of Et, whereas further enrichment with ethylene has a moderate effect. The same tendency was reported for the silane–siloxane copolymers with increasing content of flexible dimethylsiloxane units in a poly(vinyl trimethylsilane) main chain [47].

Further elongation of the alkylene fragment in the Et–VOAc main chain facilitates crystallization. Whereas the Et–VOAc copolymers are nearly amorphous up to 25 mol% of ethylene (DC ≤ 1%), the copolymers with higher ethylene content become semi-crystalline (DC = 10 and 19% for the Et–VOAc copolymers with 40 and 58% mol of Et, respectively) [48].

The gas permeability and diffusivity for the copolymers are slightly decreased accordingly (Table 4). Thus, the estimated gas permeability coefficients of O_2_ and CO_2_ for the Et–VOAc with content close to PCOAc should be not less than 10 and 70 Barrer, respectively.

Metathesis polynorbornene PN is an amorphous glassy polymer with moderate gas permeability (*P*(O_2_) = 2.3 Barrer, Table 5). The introduction of flexible-chain oligomeric blocks of COAc into the main chain of PN via the cross-metathesis reaction should increase its rotational mobility. As a result, the *T_g_* of (N–COAc)C1 copolymer is lower (Table 3) and the permeability coefficients for almost all gases are higher than the corresponding parameters for pure PN (Table 5). Moreover, higher chain mobility also results in nearly two-fold higher gas diffusivities in the copolymer relative to PN (Table 6). Therefore, it is the diffusion that mostly contributes to an increase in gas permeability, whereas the gas solubility coefficients become slightly lower in comparison with those for PN. An exception from this behavior is the carbon dioxide solubility, which is not changed or slightly increases within the measurement error range. This might be attributed to the specific interactions of a carbon dioxide molecule with a carbonyl group in the OAc moiety [56].

The influence of the post-modification of norbornene-based polymers on gas permeability has been investigated in the literature [3,5,6,7,8,9,10]. It was clearly demonstrated that the epoxidation of double bonds in PN resulted in dramatic decreases in gas diffusivity and permeability despite an increase in the glass transition temperature. Here, one can observe similar behavior—the glass transition temperature of E(N–COAc) grows by ca. 30 °C, yet remains lower than the *T_g_* for epoxidized PN (64 °C [3]). The gas permeability and diffusivity of epoxidized E(N–COAc) are considerably lower than the corresponding parameters for the initial copolymer (Table 5 and Table 6). Previously, the effect was explained by stronger interchain interactions due to the polar oxygen-containing groups that significantly restrict the mobility of penetrant molecules [3]. The same is evidently true for the E(N–COAc) copolymer. The diffusion coefficients of all gases for the (N–COAc) copolymer decrease by a factor of 13–29 upon epoxidation (Table 6). At the same time, the solubility coefficients decline by less than half (Table 6). An exception from this trend is again carbon dioxide. Its solubility coefficient increases from 1.4 to 2.1 cm^3^ (STP)·cm^−^^3^·atm^−^^1^ in accordance with the behavior of the epoxidized metathesis-based polynorbornenes [3]. The deacetylation is known to increase the glass transition temperature of ethylene–vinyl acetate copolymers [35,48,57]. Replacing the acetoxy- groups with hydroxyls allows the formation of a hydrogen bond network that reinforces interchain interactions and hinders the chain rotational mobility. Therefore, the alcoholysis of acetoxy-containing vinyl polymers should significantly reduce the gas permeability, which can be seen from the data for the vinyl alcohol–vinyl acetate (VOH–VOAc) 96/4 (% mol) copolymer and poly(vinyl alcohol) PVOH in Table 4. However, in the presence of ethylene moieties in the main chain, this effect results only in a slight decrease in gas permeability and a two-fold decrease in gas diffusivity (see the data for Et–VOH 74/26 (% mol) in Table 4 and E(N–COH)C1 in Table 5). A more pronounced effect on the gas permeability is demonstrated for the hydrolysis of acetyl cellulose, where *P*(O_2_) drops from 1.46 to 0.32 Barrer upon the increase in the substitution degree of acetyl groups from 1.75 to 2.85 [58].

As shown in Section 3.2, E(N–COH)C2 contained additional OH-groups in its chemical structure due to the opening of epoxy rings in the E(N–COAc)C2 copolymer. The additional hydroxyls intensify interchain interactions in comparison with the E(N–COH)C1 copolymer. As a result, one can observe further decreased gas permeability for E(N–COH)C2; for instance, the oxygen permeability coefficient drops by a factor of 2 from 0.14 to 0.069 Barrer (Table 5).

The hydrogenation of double bonds in metathesis-based PNs can be considered as a method to tailor their physicochemical properties. Due to the presence of substituted ordinary C–C bonds instead of the double C=C bonds or oxirane rings in the main chain, the hydrogenated PNs have higher rotational mobility in comparison with the metathesis-based or epoxidized polymers, which allows them to be packed in space more densely. Moreover, the hydrogenated H(N–COH)C2 copolymer, as well as the hydrogenated PN, [36] possesses a lower glass transition temperature and demonstrates the ability to crystallize. However, the hydrogenation does not significantly influence the gas transport. Therefore, the gas permeability coefficients for H(N–COH)C2 become slightly lower or comparable with those for epoxidized E(N–COH)C1. The interplay of two oppositely directed factors (semi-crystallinity and higher rotational mobility) might be an explanation for this behavior.

Figure 3 compares the oxygen permeability values, or alternatively, the oxygen transmission rate (OTR) values, of the norbornene-based and vinyl polymers with acetoxy and hydroxy functionalities. The ethylene–vinyl acetate copolymers with relatively low (less than 10–20% mol) fractions of ethylene are characterized by permeabilities comparable to those for the norbornene copolymers before and after post-modification. A significant barrier effect was attained via the epoxidation of main-chain double bonds in the multiblock copolymers. At the same time, the appearance of hydroxyl groups upon deacetylation led to a slight decrease in oxygen permeability, probably due to the low content (C1 contained only 6% mol of HO-groups). A similarly weak effect of the hydroxyl groups on the oxygen permeability was reported for the Et–VOH copolymers with 26% mol of vinyl alcohol units. The presence of ethylene blocks in the main chain seems to mitigate the influence of VOH groups, resulting in only a two-fold decrease in the OTR parameter (cf. the first and third columns from the right in Figure 3). Replacing the ethylene units with vinyl alcohol ones yields a VOH–VOAc copolymer (the second column from the right) with markedly better oxygen barrier characteristics.

Among our copolymers, E(N–COH)C2 contains twice as many deacetylated HO-groups (12% mol) as E(N–COH)C1 and demonstrates the lowest permeability (Figure 3, Table 5). According to the NMR and FTIR spectra, it has the largest fraction of hydroxyl and ether groups, which are formed not only as a result of deacetylation but also due to the opening of oxirane rings. The latter reaction seems to be of particular importance, which becomes clear when comparing the properties of E(N–COH)C2 and H(N–COH)C2 copolymers. Indeed, although the epoxidized E(N–COAc)C2 and hydrogenated H(N–COAc)C2 copolymers contain the same number of AcO groups, upon deacetylation the copolymer E(N–COH)C2 appears less permeable to oxygen than H(N–COH)C2, despite the crystallinity revealed by the latter polymer.

The mechanical characteristics of the copolymers are presented in Table 7. The copolymers with higher contents of norbornene units (C1) have higher elongation at break (ε) values, whereas E(N–COH)C2 demonstrates high tensile strength and Young’s modulus typical of PVOH.

Accordingly, the C2 copolymer film (Figure 4D–F) is stiffer and keeps its shape better than the C1 film (Figure 4A–C). The films of the modified copolymers decompose at temperatures higher than 300 °C. They are fairly stable during storage in air, in contrast to PN films, which quickly turn yellow.

## 4. Conclusions

In this paper, we have presented several approaches to endow metathesis polynorbornene containing main-chain double bonds with better barrier properties. The main approach included the incorporation of ethylene–vinyl acetate/vinyl alcohol blocks into the backbone of polynorbornene via its cross-metathesis with poly(5-acetoxy-1-octenylene) and further modification of the obtained multiblock (N–COAc)C copolymers by deacetylation and hydrogenation. This method provided a 15–18-fold decrease in the oxygen permeability relative to polynorbornene. The epoxidation of C=C double bonds proved to be an effective yet simpler approach to the copolymer post-modification as compared with hydrogenation and deacetylation. It was also demonstrated that the epoxy ring opening with the formation of hydroxyl and ether groups opens more opportunities to improve the barrier properties of norbornene-based polymers. We are currently working on the further development of this method.

## Data Availability

The data presented in this study are available on request from the corresponding authors.

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
