# Peer review of "Olefin-Metathesis-Derived Norbornene–Ethylene–Vinyl Acetate/Vinyl Alcohol Multiblock Copolymers: Impact of the Copolymer Structure on the Gas Permeation Properties"

_polymers, 2022, doi:10.3390/polym14030444_

Round 1

Reviewer 1 Report

The study deals with the synthesis of multiblock norbornene-ethylene-vinyl acetate and vinyl alcohol copolymers by olefin metathesis polymerization and post functionalization chemistry. The materials are novel. The synthesis routes open the way to interesting polymer materials. The authors also studied their application potential as gas permeation membranes. The work is comprehensive and thorough to a great extent. There are some points though that need further attention.

  1. The authors should provide SEC chromatograms of starting and final materials, were possible.
  2. Table 1 and related discussion: why composition of copolymers has not been varied to a wider range?
  3. It seems that during epoxidation molecular weight of the polymers decreases. Can this be bypassed in some way? Has the epoxidation reaction been optimized?
  4. p. 10, l. 393-415: XRD measurements would be helpful in determining the crystallinity of the copolymers even in this stage of the study in order to correlate it with phase and permeability properties of the materials.
  5. Can permeability measurements be performed at higher temperatures? It would be interesting to see the permeability dependence on temperature.

Author Response

Reviewer 1

  1. The authors should provide SEC chromatograms of starting and final materials, were possible.

SEC chromatograms of the starting and modified copolymers are included into Supplementary materials as Fig. S17 and S18. These figures are mentioned in the last paragraph on page 7 of the main text.

  1. Table 1 and related discussion: why composition of copolymers has not been varied to a wider range? 

We believed that it was important to identify key motifs in the influence of the copolymer composition on gas barrier and other properties. To that end, two series of copolymers were chosen, one with a higher content of norbornene blocks (C1) and another with dominating ethylene-vinyl acetate / ethylene-vinyl alcohol blocks (C2). Even in this case, the gas permeability data were presented for a fairly large series of five copolymers. Compositional effects were addressed in our other publications on multiblock copolymers (refs. 14, 20) 

  1. It seems that during epoxidation molecular weight of the polymers decreases. Can this be bypassed in some way? Has the epoxidation reaction been optimized?

 The molar mass of norbornene homo- and copolymers indeed decreases during their epoxidation. We have optimized the epoxidation conditions to obtain copolymers with highest MM in our previous study (ref. 30). This point was added to the paragraph on the epoxidation conditions in subsection 2.3 on page 4 

  1. 10, l. 393-415: XRD measurements would be helpful in determining the crystallinity of the copolymers even in this stage of the study in order to correlate it with phase and permeability properties of the materials.

This is true, so we provided the XRD intensity patterns for three of our copolymers in Figure S19. In accordance with the DSC data only one copolymer, H(NB-COH)C2, reveals crystallinity, which resembles that of poly(vinyl alcohol). We extended the last paragraph of subsection 3.2 as follows: “Here we just refer to the standard XRD patterns of H(N-COAc)C2, E(N-COH)C2, and E(N-COH)C1 copolymers shown in Figure S19. Whereas both epoxy-containing copolymers are amorphous, the hydrogenated sample reveals crystalline peaks. The positions of two main peaks located closely around 2theta = 20° give some evidence that we deal with the crystallinity of PVOH rather than PE, where the main peak is expected at 2theta > 21° [44].” A more detailed consideration will be presented elsewhere.

  1. Can permeability measurements be performed at higher temperatures? It would be interesting to see the permeability dependence on temperature.

The aim of this investigation was to characterize the transport properties of five synthesized copolymers [(N-COAc)C1, E(N-COAc)C1, E(N-COH)C1, E(N-COH)C2, H(N-COH)C2)] on six gases (He, H2, O2, N2, CO2 and CH4) at room temperature and to estimate an influence of the modification routes on the parameters. Additional determination of the thermal parameters for gas transport would have required extra measurements at four temperatures that means 4x6x6 = 144 additional experiments. We believe that the study of the temperature dependencies deserves a separate investigation.

Reviewer 2 Report

I rate the article on the assessment of the impact of changing the chemical structure of the copolymer in relation to its gas permeability, submitted for review very highly and recommend it for publication after taking into account a few minor concerns, as listed below.

  1. Please provide information on how many membranes have been tested, what is the repeatability of the permeability measurements.
  2. Have the authors analyzed the possibility of the conditioning effect of carbon dioxide, which, due to its high solubility, often plasticizes membranes and increases gas permeability, usually at the expense of selectivity?
  3. The publication contains an unjustified number of self-citations by the authors, respectively, Gringolts 18, Kudryavtsev 12, Finkelshtein 15 self-citations

Author Response

Reviewer 2

I rate the article on the assessment of the impact of changing the chemical structure of the copolymer in relation to its gas permeability, submitted for review very highly and recommend it for publication after taking into account a few minor concerns, as listed below.

  1. Please provide information on how many membranes have been tested, what is the repeatability of the permeability measurements.

 The amount of the samples for each copolymer was not less than two and the relative error of the permeability and diffusion coefficients were not higher than 10% and 20%, respectively. This information was added to the last paragraph of 2.2. Subsection on page 3. 

  1. Have the authors analyzed the possibility of the conditioning effect of carbon dioxide, which, due to its high solubility, often plasticizes membranes and increases gas permeability, usually at the expense of selectivity?

The measurements of permeability and diffusivity for carbon dioxide were carried out after four other gases (He, H2, O2, N2) and at ambient conditions, when the conditioning effect CO2 is not expected.

  1. The publication contains an unjustified number of self-citations by the authors, respectively, Gringolts 18, Kudryavtsev 12, Finkelshtein 15 self-citations 

We removed three self-citation references. Now the self-citation is within 25%. Note that we discuss remaining papers in the test, not just mention them.